Host-Microbe Biology
# Quantifying Live Microbial Load in Human Saliva Samples over Time Reveals Stable Composition and Dynamic Load

Clarisse Marotz,[a] James T. Morton,[b] Perris Navarro,[a] Joanna Coker,[a] Pedro Belda-Ferre,[a] Rob Knight,[a,c,d,e] Karsten Zengler[a,c,d]

[a]Department of Pediatrics, University of California, San Diego, La Jolla, California, USA
[b]Center for Computational Biology, Flatiron Institute, Simons Foundation, New York, New York, USA
[c]Center for Microbiome Innovation, University of California, San Diego, La Jolla, California, USA
[d]Department of Bioengineering, University of California, San Diego, La Jolla, California, USA
[e]Department of Computer Science & Engineering, University of California, San Diego, La Jolla, California, USA

**ABSTRACT** Evaluating microbial community composition through next-generation sequencing has become increasingly accessible. However, metagenomic sequencing data sets provide researchers with only a snapshot of a dynamic ecosystem and do not provide information about the total microbial number, or load, of a sample. Additionally, DNA can be detected long after a microorganism is dead, making it unsafe to assume that all microbial sequences detected in a community came from living organisms. By combining relic DNA removal by propidium monoazide (PMA) with microbial quantification with flow cytometry, we present a novel workflow to quantify live microbial load in parallel with metagenomic sequencing. We applied this method to unstimulated saliva samples, which can easily be collected longitudinally and standardized by passive collection time. We found that the number of live microorganisms detected in saliva was inversely correlated with salivary flow rate and fluctuated by an order of magnitude throughout the day in healthy individuals. In an acute perturbation experiment, alcohol-free mouthwash resulted in a massive decrease in live bacteria, which would have been missed if we did not consider dead cell signal. While removing relic DNA from saliva samples did not greatly impact the microbial composition, it did increase our resolution among samples collected over time. These results provide novel insight into the dynamic nature of host-associated microbiomes and underline the importance of applying scale-invariant tools in the analysis of next-generation sequencing data sets.

**IMPORTANCE** Human microbiomes are dynamic ecosystems often composed of hundreds of unique microbial taxa. To detect fluctuations over time in the human oral microbiome, we developed a novel workflow to quantify live microbial cells with flow cytometry in parallel with next-generation sequencing, and applied this method to over 150 unstimulated, timed saliva samples. Microbial load was inversely correlated with salivary flow rate and fluctuated by an order of magnitude within a single participant throughout the day. Removing relic DNA improved our ability to distinguish samples over time and revealed that the percentage of sequenced bacteria in a given saliva sample that are alive can range from nearly 0% up to 100% throughout a typical day. These findings highlight the dynamic ecosystem of the human oral microbiome and the benefit of removing relic DNA signals in longitudinal microbiome study designs.

**KEYWORDS** 16S sequencing, flow cytometry, longitudinal, microbial load, microbiome, propidium monoazide (PMA), relic DNA, saliva

Address correspondence to Karsten Zengler, kzengler@ucsd.edu.

How many bacteria are in a drop of drool? Counting the number of live microbes in saliva throughout an ordinary day shows massive fluctuations in microbial load.

The human oral cavity provides a distinct microbial niche and contains a complex community of microorganisms. Because saliva is easy to sample and has a relatively high biomass, it has become an increasingly popular environment for which to study

host-microbe interactions. Individuals harbor distinct salivary microbiomes (1) shaped by their environment (2) that are stable over weeks (3) and months (4). However, the effects of acute perturbations induced through daily life, including dental hygiene and eating habits, are less well understood. Determining these intricate dynamics could have major implications for cross-sectional studies impacted by these variables.

In addition to our limited knowledge about how the microbial composition of saliva changes over time, it is also unknown how the number of microorganisms, i.e., the microbial load, changes. While it is currently relatively easy to estimate the microbial composition of a given sample through sequencing, it is much more challenging to quantify the microbial load of complex communities. However, elucidating the exact number of microbial cells in and on our body is an essential component for understanding host-microbe interactions and can bias the interpretation and analysis of sequencing data (5–7). This study has important implications for contextualizing the dynamic nature of the oral microbiome that may influence our understanding of disease phenotypes.

Multiple methods have been proposed for microbial load estimation, including, but not limited to, quantitative sequencing spike-ins, extrapolation of cell number from 16S rRNA gene copy number enumeration by quantitative PCR (qPCR), and flow cytometry. Each method includes advantages and disadvantages, as previously described in detail (5, 7). The current gold standard for quantifying live microbial load from complex communities is using flow cytometry (5–9), although it is still important to keep in mind the potential presence of bacterial aggregates which may artificially deflate the cell count (10). The method presented in this study involves staining microbes with a fluorescent DNA dye and using specific gating strategies to exclude background signals without relying on cell size or density (8).

However, not all detectable DNA comes from living organisms. In 2016, Carini et al. demonstrated that in soil samples, an average of 40% of DNA originated from nonintact cells (11). The presence of extracellular DNA or DNA from nonintact cells, known as "relic DNA," has long been recognized in human microbiome samples. For example, evaluating microbial load in sputum samples from patients with cystic fibrosis without taking viability into account resulted in inaccurate clinical conclusions (12). This is due to the large amount of relic DNA in cystic fibrosis sputum samples (13). Our studies show that relic DNA can also bias interpretation of the salivary microbiome.

Treating samples with propidium monoazide (PMA) enables the removal of DNA not protected by a cell membrane (14). PMA, similar to propidium iodide, is a DNA intercalator that cannot pass intact cell membranes. Therefore, it only intercalates in relic DNA. Upon exposure to visible light, $N_2$ is photolytically cleaved from the PMA molecule resulting in a highly reactive nitrene intermediate. This reactive product quickly forms a covalent bond with the DNA to which it is intercalated, inducing a DNA break. Experiments from ethidium monoazide, a chemical cousin of PMA, demonstrated that these molecules can intercalate roughly every 10 to 80 base pairs (15), effectively excluding the resulting DNA fragments from downstream analysis. DNA fragments of this size are not recovered in typical DNA extraction kits, cannot be detected with typical fluorescent readouts, and cannot be amplified by PCR. Importantly, any additional PMA not intercalated in DNA reacts with $H_2O$ and is rendered inert following light treatment. This method has been increasingly used over the past few years to distinguish live cell DNA from relic DNA (11, 12, 16–21).

Here, we optimized a technique to quantify live microbial load from saliva samples using PMA and flow cytometry. This technique was applied to unstimulated saliva samples collected throughout a single day and in response to an acute perturbation in order to assess saliva microbial dynamics.

## RESULTS

**Approach to quantifying live microbial load and composition.** We developed a protocol to quantify live microbial load in unstimulated saliva by removing relic DNA

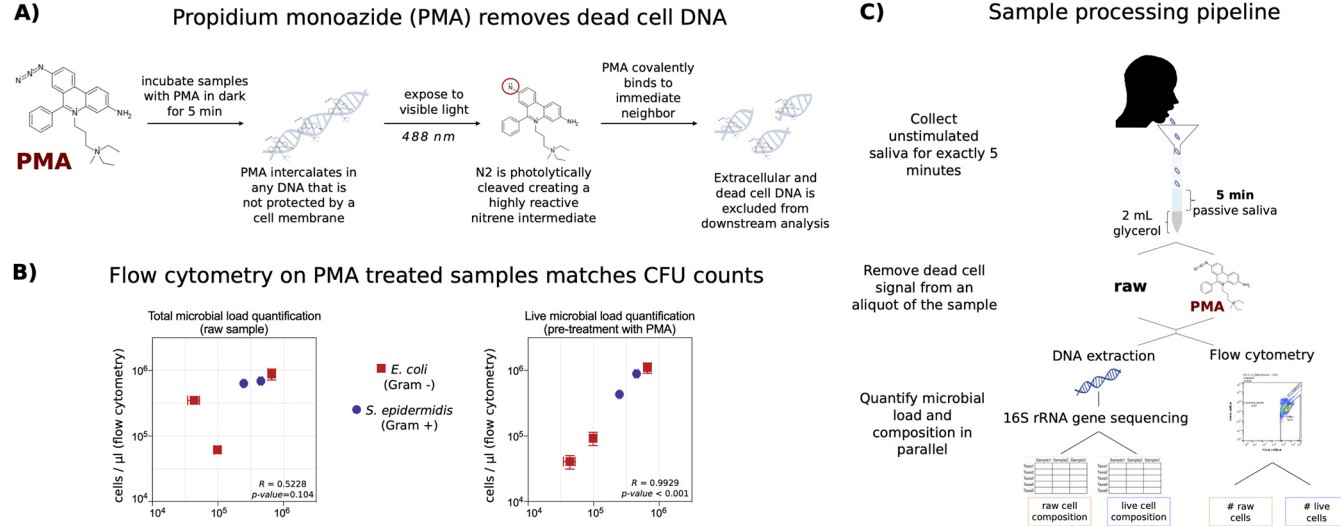

**FIG 1** Quantifying live microbial load and composition in human saliva. (A) Schematic illustrating how PMA removes DNA not protected by an intact cell membrane. (B) Flow cytometric quantification of microbial load matches CFU estimation more closely when the sample is pretreated with PMA for both a Gram-positive bacterium, *E. coli*, and a Gram-negative bacterium, *S. aureus*. Each point represents technical triplicates of each method, with the standard deviation shown for the CFU measurement (*x* axis) and flow cytometry measurement (*y* axis). (C) Schematic of sample collection and processing allowing for measurement of unstimulated salivary flow rate, live and total microbial load, and live and total microbial composition.

with PMA prior to staining for flow cytometry (Fig. 1A). Validation experiments with Gram-positive and Gram-negative bacterial cultures demonstrated a linear relationship between flow cytometry counts and classic CFU enumeration, especially when the sample was pretreated with PMA to remove relic DNA (Fig. 1B). The ability of PMA to remove dead cell signal was further validated by heat killing a fresh culture of *Escherichia coli*. Without PMA treatment, heat-killed cells were counted at similar levels to fresh cells by flow cytometry (see Fig. S1 in the supplemental material). Removing relic DNA with PMA prior to flow cytometry removed detection of these dead cells, matching the CFU counts.

With this method, we set out to determine microbial population dynamics in human saliva by quantifying live microbial load and composition throughout a single day. Specifically, we elucidated if daily perturbations (e.g., brushing teeth and eating a meal) induce ecological shifts that could negatively affect cross-sectional data sets. We collected unstimulated saliva at 9 time points throughout the day and in response to an acute perturbation (summary of participant demographics in Table 1). From each sample, we measured salivary flow rate, live microbial load, and live microbiome composition via 16S rRNA gene amplicon sequencing (Fig. 1C).

**Human saliva microbial load and viability.** In total, we processed 172 samples from 38 individuals across 2 experiments. The average unstimulated salivary flow rate was 0.48 ml per minute ($\pm$0.03 standard error of the mean), and the median flow rate was 0.38 ml per minute (range, 0.02 to 1.56), which were similar to those in previously published reports (22, 23). There was no significant difference in salivary flow rate between samples from males versus females (independent *t* test, $P = 0.398$), and there was no correlation of salivary flow rate with age (Pearson correlation coefficient, $-0.003$; $P = 0.964$). Previous studies report higher unstimulated salivary flow rate in males (23) and decreasing flow rate with increasing age (22). The lack of statistical significance for age and gender in this study may be due to the relatively low number of participants enrolled ($n = 38$) (Table 1).

The number of live microbial cells collected in 5 minutes varied by more than 3 orders of magnitude across participants and time points (Fig. 2A). The percentage of live cells (calculated by dividing the number of cells obtained from the PMA-treated sample by that of the raw sample) also fluctuated greatly throughout the day and across perturbations, ranging from <1% to 100% (Fig. 2B). Saliva samples collected

**TABLE 1** Participant demographics[a]

| Study | Treatment | Avg age | SD of age | No. of females | No. of males | No. of samples per participant | Total no. of samples |
|---|---|---|---|---|---|---|---|
| Daily dynamics | n/a[b] | 36.9 | 18.5 | 5 | 5 | 9 | 90 |
| Acute perturbation | Water | 26.9 | 5.1 | 3 | 4 | 3 | 21 |
| | Antiseptic mouthwash | 27.9 | 4.3 | 3 | 4 | 3 | 21 |
| | Alcohol-free mouthwash | 32 | 5.2 | 3 | 4 | 3 | 21 |
| | Soda | 30.1 | 3.9 | 2 | 5 | 3 | 21 |
| Total/average | | 30.76 | 7.4 | 16 | 22 | | 174 |

[a]For each experiment, the average and standard deviation of the age of the participants are listed, as well as the number of male and female participants and total number of samples collected.
[b]n/a, not applicable.

immediately after waking had a high number and percentages of live cells (red dots, Fig. 2A and B). Conversely, saliva samples collected following a rinse with alcohol-free mouthwash had low numbers and percentages of live cells (blue dots, Fig. 2A and B).

Salivary flow rate was negatively correlated with microbial concentration in saliva (Pearson correlation coefficient; $-0.326$; $P = 0.009$), and this relationship was stronger when considering only live cells (Pearson correlation coefficient, $-0.377$; $P = 0.003$) (Fig. 2C).

We also estimated bacterial concentration via qPCR of the 16S rRNA gene (see Fig. S2 in the supplemental material) (24). Quantifications by flow cytometry and qPCR were significantly correlated in both the raw (Pearson correlation coefficient, 0.272; $P = 0.011$) and PMA-treated samples (Pearson correlation coefficient, 0.333; $P = 0.002$) (Fig. S2A). These results are similar to those of a previous study by Vandeputte et al., when comparing flow cytometry to qPCR (7). This relatively low correlation coefficient is not unexpected given that qPCR and flow cytometry have different biases when determining absolute quantification, as previously discussed (5, 7). However, salivary flow rate was not significantly correlated with 16S rRNA gene copy number determined by qPCR in either the raw samples ($P = 0.753$) or PMA-treated samples ($P = 0.106$) (Fig. S2B).

In addition to microbial load, microbial composition was assessed with 16S rRNA gene amplicon sequencing. Following quality-control filtering, the median sequencing depth was 33,999 reads per sample, with a median of 147 amplicon sequence variants (ASVs) per sample. When collapsed across all participants, >99% of the entire data set belonged to 5 phyla. Within these 5 phyla, 16 genera had an average relative abundance across samples of >1% and together made up >90% of the data. Shotgun sequencing of the saliva samples collected throughout a single day ($n = 88$) provided nearly identical results, where the same top 5 phyla made up >99% and the same top 16 genera made up 89% of the taxonomic hits from Metaphlan2. These results are consistent with those of previous work looking at a range of human oral cavity sites, including tongue, mucosa, and supra- and subgingival plaque, corroborating the hypothesis that saliva microbial composition is a mix of dissociated cells across all these diverse niches of the human oral cavity (1, 25–28). Furthermore, together, these results suggest that across orders of magnitude in biomass (Fig. 2A), saliva microbial composition is highly conserved across individuals and time.

To determine the effect of PMA treatment on microbial composition, we performed differential ranking with the tool songbird (5). Songbird performs multinomial regression analysis and outputs a differential for each feature (ASV) representing its association with a given metadata category. By using PMA treatment as a variable in the model, we were able to identify taxa positively or negatively associated with PMA treatment (i.e., taxa more likely to be alive or dead, respectively). The resulting differentials were visualized with the interactive phylogenetic tool Empress (29) (Fig. 2D). This analysis revealed that taxa from the phylum *Spirochaetes* were more likely associated with raw samples, suggesting that much of the *Spirochaetes* DNA detected in these saliva samples was relic DNA. All of the *Spirochaete* ASVs in this data set were assigned to

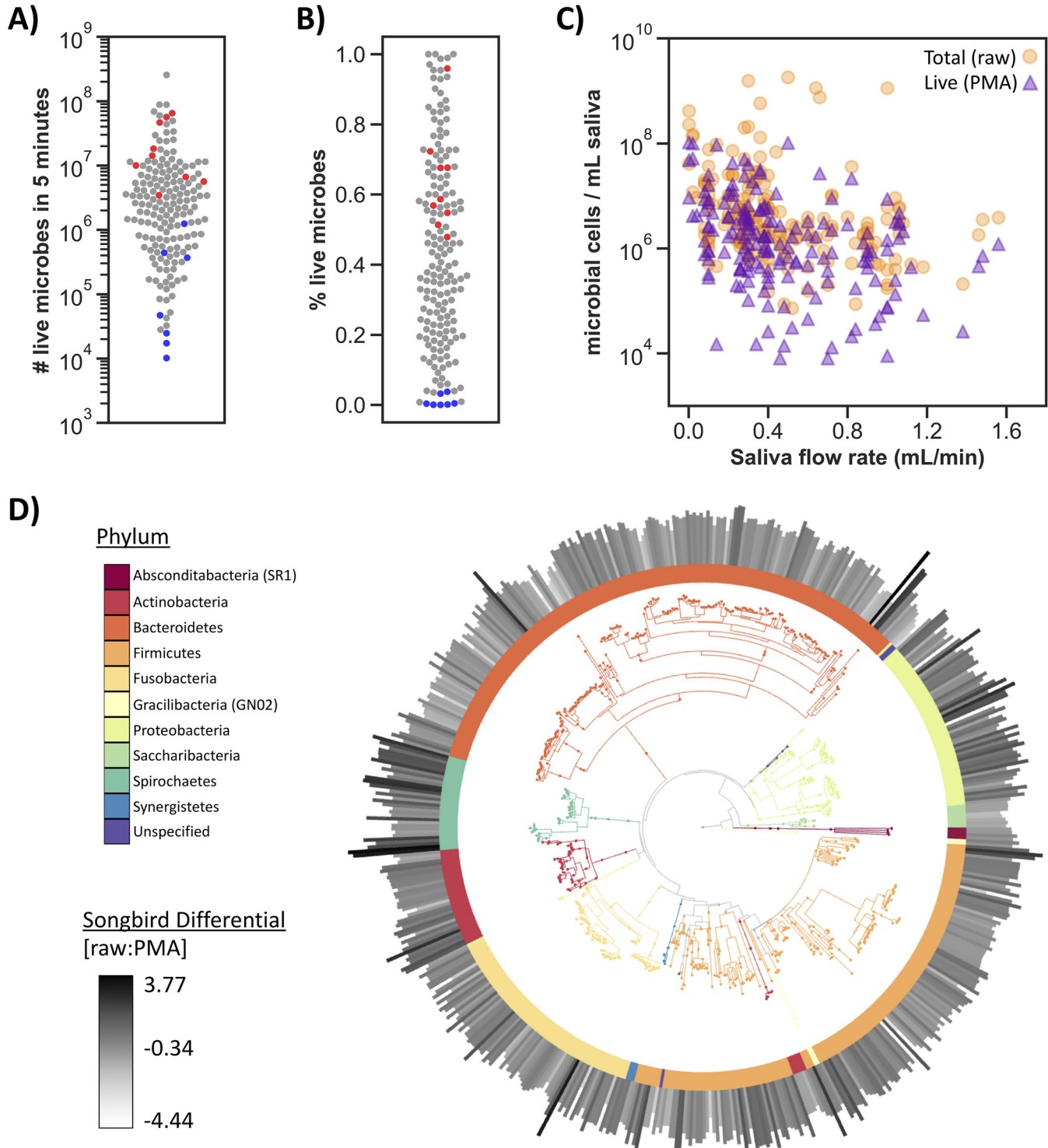

**FIG 2** Microbial load and viability vary widely across healthy participants and are negatively correlated with salivary flow rate. Samples from both the daily dynamics and acute perturbation study are shown combined here ($n$ = 172). (A) Total number of live microbial cells detected in unstimulated saliva collected for 5 minutes determined by PMA treatment and flow cytometry. (B) Percentage of live cells (number of cells after PMA treatment/number of cells detected in raw sample using flow cytometry) in saliva samples. (A, B) Red dots represent saliva samples collected immediately after waking up, and blue dots represent saliva samples collected following alcohol-free mouthwash. (C) Number of microbial cells per ml determined by flow cytometry on the $y$ axis by salivary flow rate in ml per minute on the $x$ axis. Pearson's correlation coefficient was significant for both raw samples ($r$ = −0.326, $P$ = 1.39e-5) and PMA-treated samples ($r$ = −0.377, $P$ = 3.73e-7) of log-transformed microbial concentration data (log[number of cells per ml]) and salivary flow rate (ml per min). (D) Empress plot showing the SATé-enabled phylogenetic placement (SEPP) insertion tree phylogeny with branches and the inner ring colored by phylum. The outer bar plots represent the Songbird differential for each feature, where the darker and larger the bar the more that feature is associated with raw samples compared with PMA-treated samples.

the genus *Treponema*, an anaerobic bacteria often identified in subgingival plaque and associated with periodontitis (30).

**Daily dynamics of the human saliva microbiome.** To assess the effect of daily perturbations on the saliva microbiome, we recruited 10 individuals to collect 9 saliva samples throughout the day (summarized demographics under "daily dynamics' study in Table 1; full demographics available on github repository). Participants were asked to collect unstimulated saliva first thing in the morning, before brushing their teeth or eating, then again after brushing their teeth in the morning, and then roughly every 2 hours throughout the day. We asked participants to report any additional oral hygiene events and what they ate and drank throughout the day.

Samples collected early in the morning (the first two time points collected before and after brushing teeth) tended to have lower salivary flow rates than samples collected later in the day ($P = 0.015$), in line with decades of research showing markedly lower salivary flow rates during sleep (31). The number of live cells collected in 5 minutes changed dramatically even within a given individual. When normalized to the first time point, the highest microbial load was obtained immediately upon waking (before brushing teeth), dropped after brushing teeth, and fluctuated throughout the day before falling again after the second tooth brushing (Fig. 3A).

Because many oral microbiome studies require participants to avoid eating before sample collection, we wanted to identify potential changes in saliva samples collected after eating. We compared saliva samples collected after eating to all other samples and found a significant increase in unstimulated salivary flow rate following eating ($0.537 \pm 0.039$ versus $0.43 \pm 0.030$ mean ml per minute $\pm$ SEM; independent *t* test, $P = 0.027$). However, we found no significant difference in the number of either live (Kruskal-Wallis, $P = 0.524$) or total microbial cells (Kruskal-Wallis, $P = 0.957$) collected in 5 minutes. We also evaluated the percentage of raw 16S rRNA gene amplicon sequencing reads aligning to chloroplast 16S rRNA genes, which likely came from food. Nearly half (49%) of the saliva samples collected after eating had chloroplast reads, compared with only 16% of the remaining saliva samples. Of these chloroplast-positive samples, the average relative abundance was 0.6% in samples collected after eating and only 0.04% in the remaining saliva samples. Furthermore, there were 4 times as many raw samples containing chloroplast reads as PMA-treated samples ($n = 42$ versus $n = 10$, respectively), indicating that most of the chloroplast DNA was extracellular or from nonintact chloroplasts. All chloroplast reads were filtered from the table for downstream analysis.

We next assessed $\beta$-diversity in the quality-filtered 16S rRNA gene amplicon data using robust Aitchison principal-component analysis (RPCA), which calculates distance on centered-log ratio (clr) transformed data, and is robust to sparsity and invariant to differences in total microbial load across samples (unlike other traditionally used $\beta$-diversity metrics) (32). RPCA $\beta$-diversity was mostly driven by participant, more than gender, processing method (i.e., raw versus PMA-treated samples), or whether the sample was collected after eating (Fig. 3B) (permutational multivariate analysis of variance [PERMANOVA] pseudo-F statistic by participant, 76.22; by gender, 27.43; by processing method, 0.26; by food intake, 0.67).

We next sought to determine how removing relic DNA with PMA affected microbial composition variability over time. We calculated the volume of the shape determined by an individual's samples over time in the first 3 principal coordinates of unweighted Unifrac using a convex hull analysis. The convex hull volume is the volume inside the shape of the points in three-dimensional space, so a larger convex hull volume indicates greater variability across the time points collected. Participants had greater variability over time when their samples were treated with PMA (Fig. 3C) (related *t* test, $P = 0.03$). This mirrors recent findings from Carini et al., who found that relic DNA removal with PMA enhanced their ability to distinguish longitudinally collected soil microbial communities (11). These findings suggest that relic DNA removal can improve the resolution of longitudinally collected microbial communities.

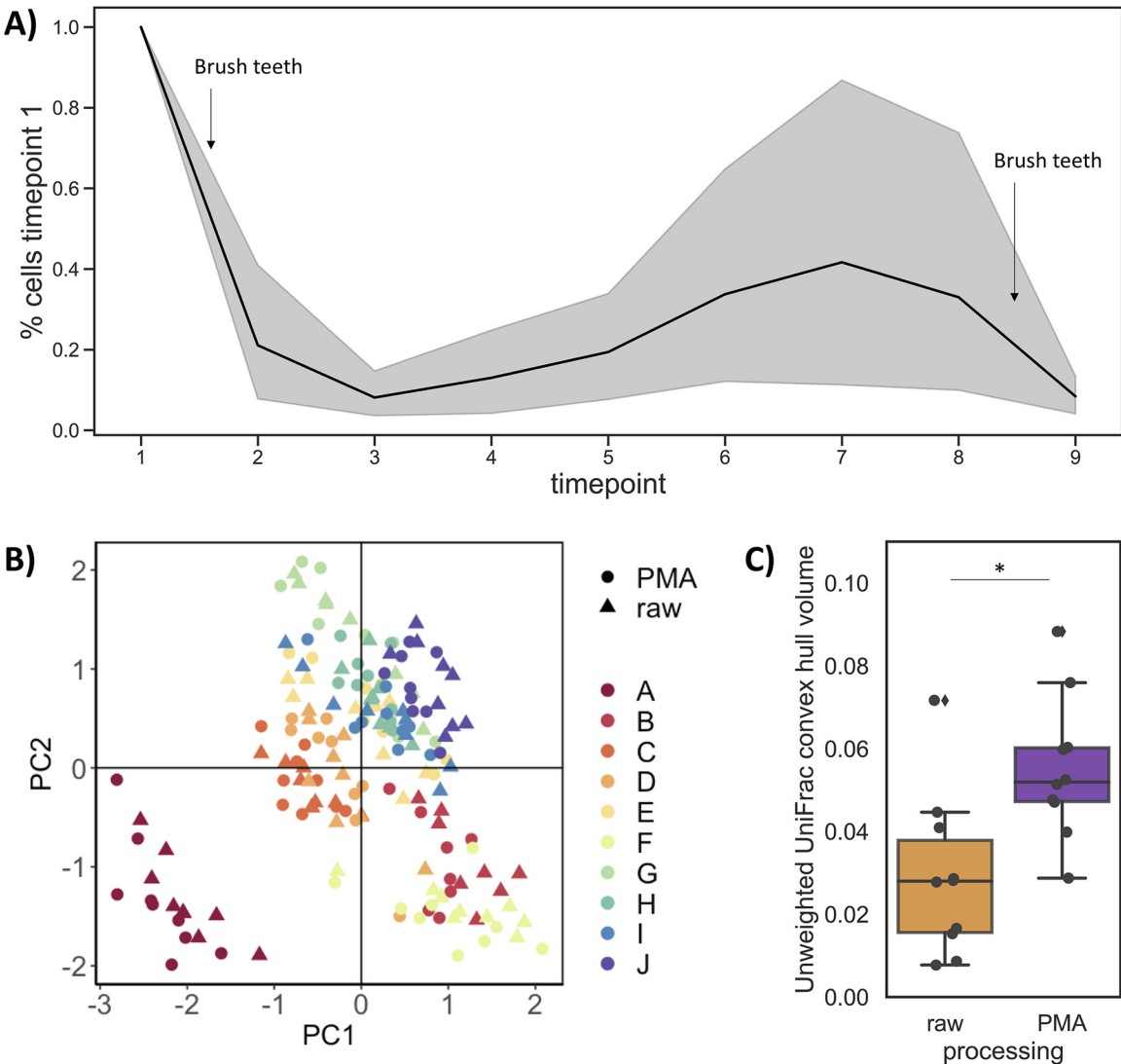

**FIG 3** Daily dynamics of the human saliva microbiome. (A) The percentage of live microbial cells collected in unstimulated saliva (for 5 minutes) normalized to the first time point; samples were collected throughout the course of a day starting from right after waking to before going to bed. (B) Robust Aitchison principal-component analysis (RPCA) reveals clustering by individual (colors) rather than treatment (shape). (C) Convex hull volume per participant on unweighted Unifrac distance shows greater microbial variation when dead cell signal was removed (PMA-processed samples). Two-sided related *t* test, *P* = 0.03.

To confirm the robustness of these results, we performed shotgun sequencing on the PMA-treated samples after processing with our lysis followed by propidium monoazide (lyPMA) protocol, which selectively depletes host DNA (33). In line with our previous host depletion experiments in saliva, the median percentage of shotgun sequencing reads aligned to the human genome was 9.8%. Metaphlan2 (34) was used to assign taxonomy to the species level, and $\beta$-diversity was assessed on the resulting table using the Aitchison distance metric. As observed in the 16S rRNA gene amplicon data, the average distance among different individuals at the same time point was greater than the average distance within an individual across time points (see Fig. S3 in the supplemental material) (boot-strapped Kruskal-Wallis, *P* < 0.001), despite large changes in microbial load. We also assessed the functional pathways present across time points and participants using HUMAnN2 and the MetaCyc database (35, 36). A total of 355 unique pathways were identified. Once again, the Aitchison distance on the pathway table revealed that samples from the same participant over time are functionally more similar than samples across different participants collected at the same

time point (Fig. S3) (Kruskal-Wallis, $P < 0.001$). The distance separating samples collected from the same individual over time versus among different individuals was largest in the 16S rRNA gene amplicon data (at the amplicon sequence variant level), smaller in the MetaPhlAn2 data (at the species level), and smallest in the HUMAnN2 data (at the functional pathway level) (Fig. S3). This finding is in line with the ecological theory of functional equivalence (or functional redundancy) underpinning a stable ecosystem (37). The microbial community in saliva is remarkably stable across daily perturbations and massive changes in microbial load, and the exact organisms in the population fluctuate more than the functional capabilities of the entire ecosystem.

**Effect of acute perturbation on saliva microbial load and composition.** To assess the effect of a controlled, acute perturbation on the saliva microbiome, we recruited 28 healthy individuals and randomly assigned them to 4 groups (summarized demographics under "acute perturbation" study in Table 1; full demographics available on github repository). Participants of each group were asked to swish for 30 seconds with either water, antiseptic mouthwash, or alcohol-free mouthwash or to drink a can of soda. Live microbial load and composition were assessed before (time point 1), 15 minutes after (time point 2), and 2 hours after (time point 3) the assigned perturbation.

To identify the effects of each treatment on microbial load, we calculated the number of nonintact ("dead") cells in each sample by subtracting the number of cells after PMA treatment (live cells) from the number of total cells detected by flow cytometry (Fig. 4A). The group that swished with water and the group that drank a soda had no significant change in microbial load across any of the time points (one-way ANOVA with Tukey's multiple-comparison test on zero-centered [log transformed] data, $P > 0.05$). In the group that swished with antiseptic mouthwash, there were significantly more dead microbial cells at time point 3 than live microbial cells at time point 2 ($P = 0.044$). This finding suggests that the active ingredients in this mouthwash (eucalyptol, thymol, methyl salicylate, and menthol) had a more delayed effect on microbial viability.

The strongest effect was in the group that swished with alcohol-free mouthwash, where there was a dramatic reduction in the number of live microbial cells directly after treatment ($P = 0.001$). Strikingly, live microbial load returned to baseline levels within 2 hours (Fig. 4A). Representative flow cytometry data on the final gated bacteria at time point 1 (before mouthwash), time point 2 (15 minutes after mouthwash), and time point 3 (2 hours after mouthwash) show the dramatic impact on microbial load induced by rinsing with alcohol-free mouthwash (Fig. 4B).

Throughout these massive fluctuations in microbial load, there was no correlation between alpha-diversity (measured by Faith's phylogenetic diversity) and the number of microbial cells (Pearson correlation coefficient $P > 0.2$ for both live and total cell counts) (Fig. 4C). Again, this result suggests that although there are major fluctuations in microbial load, the overall saliva microbial composition is relatively stable.

This finding is supported by the fact that $\beta$-diversity was largely driven by participant (RPCA PERMANOVA pseudo-F statistic by participant, 24.0; by gender, 6.9; by treatment group, 3.4; by processing method, 1.1). We were interested to see if any of the four treatments groups caused changes in bacterial composition. To assess this aspect, we calculated the pairwise unweighted UniFrac distance among samples collected from a single participant across time points (calculating convex hull volume requires more samples per participant). In the raw samples, pairwise unweighted UniFrac distance was higher in the alcohol-free mouthwash and soda groups than in the antiseptic mouthwash group (Fig. 4D). When relic DNA was first removed with PMA, these groups were also significantly different from the water group (Fig. 4D). These results highlight the ability of PMA to improve our ability to distinguish groups over time.

Furthermore, mirroring our observation that microbial load returns to baseline levels within 2 hours, we found that in the group that swished with alcohol-free mouthwash live microbial composition at time point 1 was more similar to time point 3 than to time point 2 (bootstrapped Kruskal-Wallis, $P < 0.001$) (see Fig. S4 in the

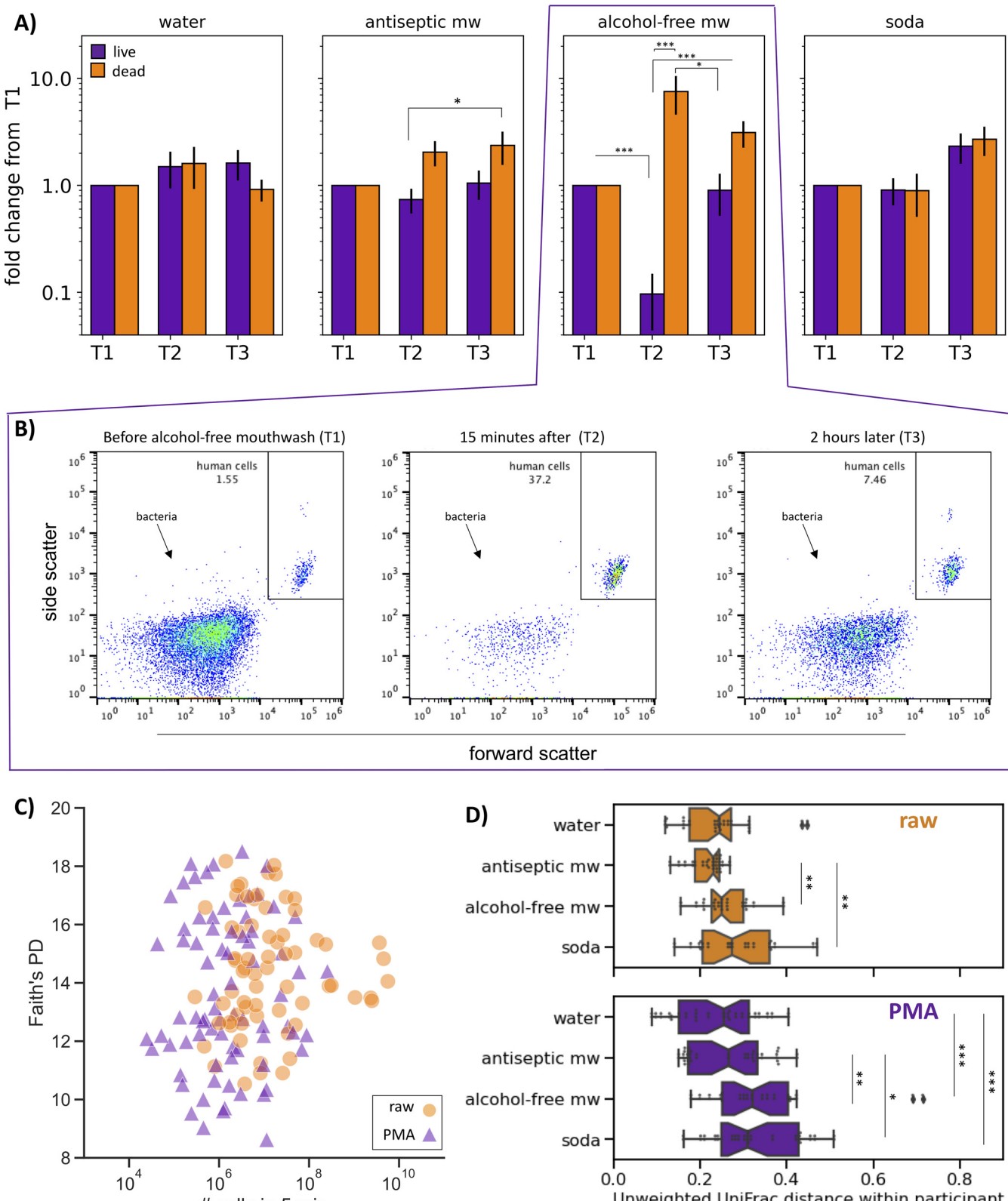

**FIG 4** The effect of acute perturbation on live and dead microbial load and composition. (A) Change in microbial load following treatment over time. The *y* axis shows percentage of live (purple) and dead (orange) microbial cells detected by flow cytometry normalized per-individual to the first time point (*n* = 7 per group). Significance represents one-way ANOVA with Tukey's multiple-comparison test; *, $P \leq 0.05$, **, $P \leq 0.01$, ***, $P \leq 0.001$. (B) Representative flow cytometry scatterplots of PMA-treated samples from a single individual before (T1) immediately after (T2) and 2 hours after (T3) rinsing with alcohol-free mouthwash. (C) Each dot represents a saliva sample processed with (purple) or without (orange) PMA to remove relic DNA (revealing live and total

## DISCUSSION

Here, we present a method to perform parallel live microbial load quantification and sequencing. This method combines the following two established protocols: PMA treatment to remove relic DNA and microbial quantification via flow cytometry. PMA treatment can be performed without any specialty lab equipment, and the flow cytometry protocol requires a standard flow cytometer equipped with a 488-nm laser and standard detectors. We applied this method to unstimulated saliva samples collected longitudinally.

To our knowledge, this study contains the first longitudinal human microbiome data set with matched 16S rRNA gene amplicon sequencing and flow cytometry quantification, each with total and live (intact) cell evaluation. In addition to providing unprecedented information about the dynamics of the human saliva microbiome, we expect that this data set will be useful to the community for evaluating novel algorithms (5, 9, 38, 39) to describe and predict ecological shifts in human microbiome samples.

Using this data set, we demonstrated an inverse relationship between salivary flow rate and microbial load. This relationship could help explain why decreased salivary flow rate is often associated with microbial-derived periodontal diseases, such as caries and gingivitis (40–42). Interestingly, microbial concentration is also negatively correlated with water content in stool samples (7). Together, these findings demonstrate the ability of the human body to modulate microbial concentration at mucosal sites.

We found that live microbial load in saliva fluctuates by orders of magnitude throughout a typical day. Despite this fluctuation, taxonomic composition is remarkably consistent across time and within individuals, with more than 90% of all 16S rRNA gene sequences coming from 16 genera across 5 phyla, similar to findings from previous studies (25–28).

The compositional effects of removing relic DNA with PMA were relatively subtle, as this processing step affected taxonomic composition much less than the effects of time or different individuals. However, we did identify that features from the genus *Treponema* were more associated with raw samples, indicating that *Treponema* detected in saliva is likely already dead. This result is unsurprising given that *Treponema* sp. is an anaerobic periodontal pathogen with high proteolytic capacity allowing it to dig into compromised gums (43). Although clinical periodontal status was not collected from participants in this study, they all self-reported as healthy, and it would be interesting to compare the status of *Treponema* signal in saliva from individuals with and without clinically evaluated periodontitis. We hope that this proof-of-concept study will allow future experiments to account for absolute microbial load and live/dead microbial status with clinical outcomes in mind.

In addition to changes in microbial composition, this study identified a clear example (alcohol-free mouthwash) where the assumption that all DNA comes from living organisms can lead to false conclusions; without taking relic DNA into account, it would appear that microbial load increases after alcohol-free mouthwash, although this treatment decimated the live microbial cell population in saliva. This result suggests that while PMA treatment may not greatly influence the results of cross-sectional oral microbiome data

**FIG 4** Legend (Continued)
microbial populations, respectively). The *y* axis is Faith's phylogenetic diversity rarefied to 10,000 sequences, and the *x* axis is the number of microbial cells collected in 5-minute unstimulated saliva detected by flow cytometry (Pearson's correlation coefficient for both live and total populations, $P > 0.05$). (D) Unweighted UniFrac distance on 16S rRNA gene amplicon sequencing data between samples from the same individual over time plotted by treatment group. Overall, the distance among samples treated with PMA (purple, bottom graph) were more dissimilar than the distances between the same samples not treated with PMA (orange, top graph) (bootstrapped Kruskal-Wallis, $P \leq 0.001$). In the raw samples, there was no significant difference among the treatment groups (bootstrapped Kruskal-Wallis, $P \geq 0.05$). In the PMA-treated samples, participants that swished with alcohol-free mouthwash or drank a soda had significantly greater variation over time than the group that swished with water (bootstrapped Kruskal-Wallis; *, $P \leq 0.05$, **, $P \leq 0.01$, ***, $P \leq 0.001$).

sets, it could add greater resolution to longitudinal studies and potentially reveal patterns detectable only in the absence of relic DNA. Interestingly, these findings mirror recent results in a soil microbiome data set. Carini et al., found that relic DNA removal improved their ability to detect changes in underground microbial communities over time (11). This technique may also greatly influence the analysis of samples containing polymicrobial biofilms, such as dental plaque, which contain large amounts of relic (or extracellular) DNA (44, 45). Together, these findings highlight how relic DNA can obscure changes in community composition across a broad range of microbial environments.

Participants in saliva microbiome experiments are often asked to avoid eating or oral hygiene anywhere from 2 hours to 24 hours before sampling, based on the assumption that these daily perturbations could influence the oral microbiome. Because of the detailed metadata collected in our longitudinal experiment, we were able to determine the effect of eating and oral hygiene on saliva microbial load and composition. Salivary flow rate was significantly higher in samples collected after eating, but the total number of cells detected in 5 minutes remained unchanged. We also found an increase in the presence and abundance of chloroplast sequences in saliva samples collected after eating, which are presumably from food sources. Chloroplast and mitochondrial reads are typically filtered out of 16S rRNA gene amplicon data sets from human microbiomes, and after removing these sequences, we found no detectable alteration in salivary microbiome composition after eating. Although previous work has identified acute effects on dental plaque biofilm following eating (46), these data suggest that the acute effect of eating has a minimal impact on the saliva microbiome.

While oral hygiene significantly reduced microbial load, the composition remained relatively unperturbed when using analytical tools that account for compositional bias (5). The decrease in microbial load was detectable after brushing teeth in both the raw and PMA-treated samples, suggesting the physical removal of bacterial cells. In contrast, swishing with alcohol-free mouthwash permeabilized cells but did not physically remove them since the decrease in microbial load was detectable only after removing dead cell signal with PMA. We were surprised that the alcohol-free mouthwash had such a dramatic effect on microbial viability but antiseptic mouthwash did not. This difference may be due to the presence of natural and artificial compounds with antimicrobial activity found in the alcohol-free mouthwash used in this study (e.g., poloxamer 407, thymol, eucalyptol, menthol, salicylate, sodium lauryl sulfate, and benzoic acid). This finding highlights the utility of using PMA to remove dead cell signal.

Together, these results showcase the remarkable resilience of the salivary microbiome and highlight the potential of the human salivary microbiome as an easily accessible model to probe complex community organization and response. Despite dramatic perturbations in the number of microbes in saliva, taxonomic compositions remain relatively stable and are strongly host specific, which lends credence to cross-sectional saliva microbiome studies when samples are collected at different times of day.

In conclusion, we present a novel method to assess live microbial composition and load. We identified a negative correlation between salivary flow rate and microbial load, which has implications for oral health. The numbers of microbes in human saliva varied by orders of magnitude throughout a single day and in response to daily perturbations. Furthermore, by removing relic DNA signal with PMA treatment prior to quantification, we were able to identify the antimicrobial effect of alcohol-free mouthwash which was missed when quantifying the raw sample. These findings highlight the importance of using analytical tools that account for compositional bias, which is exacerbated by comparing samples with different microbial loads (5–7, 47, 48). Despite these changes in absolute abundance, we found that microbial composition is extremely stable and host specific and is not significantly affected by meals or oral hygiene. These results support the use of scale-invariant tools in the analysis of cross-sectional data sets, provide a proof of concept for live microbial load assessment in human microbiome samples, and demonstrate the ability of relic DNA removal to increase the resolution of longitudinal studies.

## MATERIALS AND METHODS

**Participant recruitment and saliva collection.** Self-described healthy volunteers were recruited in accordance with institution review board (IRB) number 150275. Participant demographics are detailed in the supplementary metadata files and summarized in Table 1.

Each participant was given a kit with a funnel (catalog number F490-2; Simport) and a 15-ml collection tube (catalog number 352057; Corning) loaded with 2 ml 40% sterile glycerol for each time point to be collected. Glycerol was used to preserve the bacterial cells from freezing for downstream PMA treatment (33). The final concentration of glycerol varied from 8% to 38% but did not correlate with microbial viability (see Fig. S5 in the supplemental material) (Pearson correlation coefficient, 0.36), suggesting that variability in final glycerol concentration did not influence our ability to detect live cells in cryopreserved samples. The instructions for unstimulated saliva collection and salivary flow measurement were adapted from Navesh et al. (49). Briefly, participants were asked to find a comfortable seat, set a timer for 5 minutes, swallow, and start the timer. With the head tilted down slightly and lips against the edge of a food-grade funnel, they were asked to relax the jaw, mouth, and tongue, allowing saliva to pool and eventually drain into the tube. Participants were asked not to swallow during the 5-minute collection. After collection, the lids of the tubes were closed, and the funnel was disposed. The tubes were gently inverted 10 times to mix the saliva and glycerol and placed at −20°C. Samples were brought to the lab the following day on wet ice and stored at −20°C until further processing.

In the daily dynamics experiment, 10 individuals were recruited to collect unstimulated saliva throughout a single day. Participants were asked to collect saliva on their own (as detailed above) at 9 time points throughout the day, as follows: immediately upon waking (before eating or brushing teeth), soon after brushing teeth, and roughly every 2 to 3 hours throughout the day. Participants were asked to record the time of each saliva collection and whether they ate or performed oral hygiene since the last sample collection. Of the 90 samples, 2 were excluded because of missing sample collections, totaling 88 samples.

In the acute perturbation experiment, 28 individuals were recruited to collect unstimulated saliva before, 15 minutes after, and 2 hours after an acute treatment. Participants were asked not to eat or drink (except water) for 1 hour prior to the first collection. Each participant was assigned to 1 of 4 groups in a blind manner; they rinsed with bottled water (Simple Truth water, Kroger; pH 7.6), rinsed with mouthwash (Listerine antiseptic Cool Mint), rinsed with alcohol-free mouthwash (Listerine zero alcohol Cool Mint), or drank a 12-ounce can of Coca-Cola soda. For the rinsing samples, participants were asked to swish a 20-ml solution in their mouths for exactly 30 seconds and to refrain from gargling in the back of their mouths. For the soda samples, participants were asked to drink the 12-ounce can within 10 minutes.

**Bacterial culturing for proof-of-concept experiments.** *Escherichia coli* and *Staphylococcus epidermidis* cultures were used for proof-of-concept experiments to validate the quantitative flow cytometry protocol. Cultures of *S. epidermidis* or *E. coli* were grown overnight in tryptic soy broth (TSB) at 37°C. Bacterial cultures were pelleted by centrifugation at 8,000 × $g$ for 5 minutes and resuspended in 1× sterile phosphate-buffered saline (PBS). Resuspended cultures were run across 5-$\mu$m syringe filters (Sartorius Stedim Biotech GmbH) and diluted between 100- and 10,000-fold in PBS. Samples treated with PMA were mixed with 10 $\mu$M PMA, vortexed briefly, incubated at room temperature protected from light for 5 minutes, and then exposed to light by laying on ice <20 cm from a benchtop fluorescent light bulb for 25 minutes. Both raw and PMA samples were stained with 0.1× SYBR green (SYBR green I nucleic acid gel stain; Invitrogen) and incubated in the dark for 15 minutes at 37°C. Flow cytometry was performed using the Sony SH800 and AccuCount fluorescent particles as described below. CFU measurements were performed by plating triplicate 10-$\mu$l drops of 10-fold serial dilutions of the bacterial culture onto TSB agar plates and incubating the plates at 37°C overnight. The following day, dilutions containing between 10 and 50 colonies per drop were counted. For the heat-killed experiment (Fig. S1), overnight *E. coli* cultures were heated to 65°C for 10 min and analyzed as described above.

**Flow cytometry.** Cryopreserved, unstimulated saliva samples were thawed on ice and centrifuged at 3,000 × $g$ for 1 minute to remove any bubbles and allow for accurate volume assessment. The volume (to the closest 0.1 ml) and the weight of each sample were recorded.

Unstimulated saliva samples were diluted 10-fold with sterile, 1× PBS. To remove human cells and salivary debris, samples were filtered using a sterile 5-$\mu$m syringe filter (Sartorius Stedim Biotech GmbH). Relic DNA was removed from an aliquot of each sample; a final concentration of 10 $\mu$M PMA (Biotium) was added to 1 ml of diluted saliva, vortexed briefly, and incubated at room temperature protected from light for 5 minutes. Samples were lain horizontally on ice <20 cm away from a benchtop fluorescent light bulb for at least 25 minutes and vortexed briefly every ~5 to 10 minutes.

An aliquot of diluted, filtered saliva (raw) and the PMA-treated aliquot were then stained in parallel with SYBR green for detection by flow cytometry. A total of 5 $\mu$l 20× SYBR green (SYBR green I nucleic acid gel stain; Invitrogen) was added to 1 ml of the microbial suspension (0.1× final concentration) and incubated in the dark for 15 minutes at 37°C. Finally, 50-$\mu$l AccuCount fluorescent particles (ACFP-70-10; Spherotech) was added for quantification of microbial load. Samples were processed on a SH800 cell sorter (Sony Biotechnology) using a 100-$\mu$m chip with the threshold set on FL1 at 0.06% and gain settings set as follows: FSC = 4, BSC = 25%, FL1 = 43%, and FL4 = 50%. For the acute perturbation experiment, the threshold was increased to FL1 of 0.58% to reduce the background signal that was observed in a small number of samples. The gating strategy was adapted from Props et al. (8) and an example is shown in Fig. S6 in the supplemental material. Briefly, fluorescent microbial cells were gated from background on a FL1-FL4 density plot. Aggregates were excluded by taking the linear fraction on a graph of height versus width of the FL1 signal, and the remaining background was removed by eliminating large events detected on a forward scatter versus backward scatter density plot. Negative controls (sterile PBS

stained identically to samples) were run between each sample set to exclude cross-contamination. The final calculation of cells per $\mu$l was performed per the manufacturer's instructions of the AccuCount counting beads and taking into account the dilution factor from the glycerol preservative.

**Microbial load estimation with qPCR.** To create a standard ladder for 16S rRNA gene copy number extrapolation, genomic DNA (gDNA) from *Escherichia coli* was amplified with the Kapa HiFi HotStart ReadyMix PCR kit (catalog number KK2602) using the Bakt 341F-805R 16S rRNA gene amplicon primers (50) (Bakt_341f, 5'-CCTACGGGNGGCWGCAG-3'; and Bakt_805R, 5'-GACTACHVGGGTATCTAATCC-3'). Amplification was performed in triplicate 20-$\mu$l reaction mixtures containing 10-$\mu$l Kapa Readymix, 1-$\mu$l primer mix containing 5 $\mu$M forward and reverse primers, 2-$\mu$l gDNA, and 7 $\mu$l H$_2$O. The PCR mix was cycled through the following temperatures on a Bio-Rad CFX instrument: 95°C for 5 min, 30 cycles of 95°C for 30 sec and 60°C for 30 sec, and then 4°C hold. The PCR product was run on a 1.5% agarose gel, and the band was excised and purified using the QIAquick gel extraction kit. Amplicon concentration was quantified in triplicate using Thermo Fisher Qubit fluorometric quantitation.

To estimate microbial load, sample gDNA extracted from 200-$\mu$l saliva was evaluated in triplicate with Kapa universal qPCR master mix (catalo number KK4828) using the Bakt 341F-805R primers listed above. Amplification was performed in triplicate 10-$\mu$l reactions with each containing 5-$\mu$l Kapa master-mix, 0.5-$\mu$l primer mix containing 5 $\mu$M forward and reverse primers, 2-$\mu$l gDNA, and 2.5 $\mu$l H$_2$O. The PCR mix was cycled through the following temperatures on a Roche LightCycler 480 instrument: 95°C for 5 min, 40 cycles of 95°C for 30 sec and 60°C for 30 sec, and then 4°C hold. Triplicate, 10-fold serial dilutions of the standard ladder described above ranging from 1.3 to 1.3E + 07 copies were run in parallel and used to extrapolate the number of 16S rRNA gene copies in the saliva samples (24).

**PMA treatment and DNA extraction.** For the daily dynamics cycle experiment, 500-$\mu$l aliquots of each saliva sample was set aside for standard DNA extraction. One-milliliter aliquots were used for lyPMA treatment, which depletes human DNA and dead microbial signal (33); samples were centrifuged at 10,000 × *g* for 8 minutes. The supernatant was removed, and the cell pellet was resuspended in 200 $\mu$l sterile, pure, H$_2$O and allowed to sit at room temperature for 5 minutes to selectively lyse human cells. PMA was added to a final concentration of 10 $\mu$M, vortexed, and left in the dark at room temperature for 5 minutes. Samples were lain horizontally on ice <20 cm away from a benchtop fluorescent light bulb for at least 25 minutes and vortexed briefly every ~5 to 10 minutes. The raw and PMA-treated aliquots were frozen at −20°C until DNA extraction with the Qiagen PowerSoil MagAttract DNA kit as previously described (51). A subset of lyPMA samples with low microbial load failed in sequencing, and the PMA treatment was repeated on samples with up to 1.5 ml of unstimulated saliva. Quality-control analysis of the sequencing data showed that these reprocessed samples were similar to the matched raw samples, and this higher volume was used in the follow-up acute perturbation experiment.

For the acute perturbation experiment, samples were processed in a 96-well plate format to increase sample throughput. First, samples were vortexed thoroughly (15 seconds) and 1.5 ml was transferred into a 96-deep-well plate. Cells were pelleted by centrifugation at 3,200 × *g* for 15 minutes. One-milliliter of supernatant was removed with a multichannel pipette, and then a single channel pipette was used to remove the remaining supernatant. Pellets were resuspended in 200 $\mu$l sterile, DNase free H$_2$O by pipet-ting up and down ~12 times and leaving the pellets to sit at room temperature for 5 minutes to allow for selective lysis of mammalian cells. The plate was then briefly centrifuged (~1,000 × *g* for 1 minute), and 200 $\mu$l of each sample was transferred to a 96-well round-bottom plate (catalog number 650101; Greiner Bio-One). A total of 2 $\mu$l of 1 mM propidium monoazide (PMA) was added to each sample for a final concentration of 10 $\mu$M. The plate was covered with a transparent seal (catalog number B0443499; Bio-Rad microseal "B' seal), vortexed briefly, and left to sit at room temperature in the dark for 5 minutes. The plate was then placed on ice <20 cm away from a fluorescent bulb light source. Light exposure took place for ~30 minutes with vortexing every ~5 to 10 minutes. Immediately after light exposure, the samples were briefly centrifuged (~1,000 × *g* for 1 minute) and transferred to a Qiagen Magattract 96-well plate and extracted as previously described (51).

**16S rRNA gene amplicon sequencing and analysis. Sequencing.** gDNA was processed for 16S rRNA gene amplicon sequencing using primers against the V4 region of the 16S rRNA gene 515F-806R according to the Earth Microbiome Project protocol (52). The pooled library was sequenced on the Illumina MiSeq platform with a paired-end 150 V2 kit.

**Quality control.** Data were processed using QIIME 2 (53). Demultiplexed sequences were quality fil-tered for q-score with default settings and processed with deblur (54) trimmed to 150 bp. Samples with less than 1,000 quality-filtered sequences were dropped from downstream analysis. Following quality control, 6 samples dropped out of the acute perturbation experiment (*n* = 78). In order to remove mito-chondrial and chloroplast reads, sequences were aligned to the Greengenes database (55), and all sequences aligning to mitochondrial and chloroplast reads were filtered out using the taxonomy-based filtering command "qiime taxa filter-table."

**Taxonomic assignment.** Microbial taxonomy was assigned to the quality-filtered sequences using the Human Oral Microbiome Database v15.1 (HOMD) (56) with the "qiime feature-classifier classify-sklearn" command on a scikit-learn classifier created from the HOMD (56).

**α- and β-Diversity analysis.** QIIME 2 was used to calculate α- and β-diversity using the "qiime diver-sity" commands. For α-diversity, an α-rarefaction curve was generated to determine the appropriate subsampling sequencing depth (see Fig. S7 in the supplemental material).

**Shotgun sequencing and analysis.** For the daily dynamics study, gDNA from lyPMA-treated sam-ples (33) was quantified with the Quant-iT PicoGreen double-stranded DNA (dsDNA) assay kit (ThermoFisher Scientific), and 1 ng of input DNA was used in a 1:10 miniaturized Kapa HyperPlus proto-col. For samples with less than 1-ng DNA, a maximum volume of 3.5-$\mu$l input was used. Equimolar

amounts of each sample was pooled, and the library was size selected for fragments between 300 and 700 base pairs on a Sage Science PippinHT system. The pooled library was sequenced as a paired-end 150-cycle run on an Illumina HiSeq2500 v2 run at the UCSD IGM Genomics Center. Demultiplexed reads were quality filtered with Trim galore v0.4.2 (57). Reads aligning to the host genome (GRCh38.p7) were identified using Bowtie 2 v2.3.0 (58) with parameters set by the flag -very-sensitive-local and removed from the analysis (median percentage of host aligned reads 9.8%, similar to previously reported numbers using the lyPMA method [33]). Samples with less than 10,000 microbial reads were excluded from the analysis, leaving a total of 71 samples with a median of 348,242 quality-filtered microbial reads per sample. Taxonomic assignment was performed with MetaPhlAn v2.0 (34) using the default parameters. Functional assignment was performed with HUMAnN2 (35) using the default parameters.

**Statistical analyses.** All quality-filtered tables and the code written to produce the figures and statistical tests presented in the manuscript can be viewed and reproduced using Jupyter iPython Notebooks through github online at https://github.com/knightlab-analyses/Saliva_quantification_study.

For significance testing based on distances from sequencing data, a permutation test was used (perm_test.py in github repository). This test was chosen since univariate statistical tests often assume that observations are independently and identically distributed, which is not the case with distance calculations. Similar to PERMANOVA, the group labels were shuffled, and a Kruskal-Wallis test was applied. $P$ values were calculated by $[\#(K > Kp) + 1]/(\text{number of permutations} + 1)$ where # is the number of times that $K > Kp$ over all of the permutations, $K$ is the Kruskal-Wallis statistic on the original statistic, and $Kp$ is the Kruskal-Wallis statistic computed from the permuted grouping. A total of 1,000 permutations were used for the permutation test.

**Ethics approval and consent to participate.** Self-described healthy volunteers were recruited under IRB number 150275, approved by the UCSD Human Research Protections Program under federal-wide assurance number FWA00004495.

**Consent for publication.** Consent from each participant was obtained in accordance with IRB number 150275.

**Data availability.** All raw sequencing data collected in this study are available through Qiita (59) under study identifier (ID) 11896 (daily dynamics) and 11899 (acute perturbation) and the European Nucleotide Archive under accession number ERP111447 (daily dynamics) and accession number ERP117149 (acute perturbation).

Additionally, all metadata and the processed, quality-filtered sequencing tables as described in the Materials and Methods are available in the supplementary tables, and all analyses for figure generation and statistical analyses are publicly available through github online at https://github.com/knightlab-analyses/Saliva_quantification_study.

## SUPPLEMENTAL MATERIAL

Supplemental material is available online only.

**FIG S1**, TIF file, 2.5 MB.
**FIG S2**, JPG file, 0.4 MB.
**FIG S3**, JPG file, 0.2 MB.
**FIG S4**, TIF file, 1.5 MB.
**FIG S5**, TIF file, 2.8 MB.
**FIG S6**, JPG file, 0.3 MB.
**FIG S7**, JPG file, 0.1 MB.

## ACKNOWLEDGMENTS

We thank Cristal Zuniga for assistance with data visualization; and Julia Toronczak, MacKenzie Bryant, and Elaine Guo for assistance with processing the flow cytometry experiments.

We declare that we have no competing interests.

This work was funded in part by the Army Research Office under grant number W911NF1810158. C.M. was funded by NIDCR NRSA F31 fellowship 1F31DE028478-01.

C.M., R.K., and K.Z. designed the experiment. C.M. optimized the protocol, ran the experiment, analyzed the data, and wrote the manuscript. J.T.M. assisted in the analysis. P.N. performed proof-of-concept experiments. J.C. and P.B.-F. assisted in writing and editing the manuscript.

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
