## [Reviewer comments · mSystems]

Quantifying live microbial load in human saliva samples over time reveals stable composition and dynamic load

Clarisse (Lisa) Marotz, Jamie Morton, Perris Navarro, Joanna Coker, Pedro Belda-Ferre, Rob Knight, and Karsten Zengler

Corresponding Author(s): Karsten Zengler, University of California, San Diego

Review Timeline:

Submission Date:	November 11, 2020
Editorial Decision:	January 11, 2021
Revision Received:	January 15, 2021
Accepted:	January 23, 2021

Editor: Holly Bik

Reviewer(s): The reviewers have opted to remain anonymous.

Transaction Report:

DOI: <https://doi.org/10.1128/mSystems.01182-20>

January 11, 2021

Dr. Karsten Zengler
University of California, San Diego
Bioengineering
9500 Gilman Drive
La Jolla

Re: mSystems01182-20 (Quantifying live microbial load in human saliva samples over time reveals stable composition and dynamic load)

Dear Dr. Karsten Zengler:

Both reviewers were supportive of the study scope and results, and have provided a number of minor comments to further strengthen the manuscript.

Below you will find the comments of the reviewers.

To submit your modified manuscript, log onto the eJP submission site at <https://msystems.msubmit.net/cgi-bin/main.plex>. If you cannot remember your password, click the "Can't remember your password?" link and follow the instructions on the screen. Go to Author Tasks and click the appropriate manuscript title to begin the resubmission process. The information that you entered when you first submitted the paper will be displayed. Please update the information as necessary. Provide (1) point-by-point responses to the issues raised by the reviewers as file type "Response to Reviewers," not in your cover letter, and (2) a PDF file that indicates the changes from the original submission (by highlighting or underlining the changes) as file type "Marked Up Manuscript - For Review Only."

Due to the SARS-CoV-2 pandemic, our typical 60 day deadline for revisions will not be applied. I hope that you will be able to submit a revised manuscript soon, but want to reassure you that the journal will be flexible in terms of timing, particularly if experimental revisions are needed. When you are ready to resubmit, please know that our staff and Editors are working remotely and handling submissions without delay. If you do not wish to modify the manuscript and prefer to submit it to another journal, please notify me of your decision immediately so that the manuscript may be formally withdrawn from consideration by mSystems.

Sincerely,

Holly Bik

Editor, mSystems

Journals Department
Reviewer comments:

Reviewer #1 (Comments for the Author):

Though there are already a number of studies published out there to validate the stability of the human oral microbiome, this study brings in an interesting perspective with the influence of live cells and total microbial load. These methods should hopefully get adopted by more researchers for better validation of results.

I only have some minor comments:

Line 120: Fig 1 legend text: should be "Gram positive" and "Gram negative"

Lines 137 & 139: Should be Fig 2A and Fig 2B; also anywhere else in the document to maintain uniformity with the figure notations

Line 162, 197, 275: I don't see any supplementary tables submitted in the files section. It would be very interesting and also necessary to have this data made available - particularly the metadata details and the microbial composition files.

Line 200: Were there any patterns observed in the salivary loads or composition wrt influence of the type of diet consumed?

Lines 439 & 440: temperature should be "-20 degrees celsius"

Line 536: "...frozen at -20 C" (minus is missing)

Line 542: Any particular rationale for the two sets of samples having been processed differently? Might be worth mentioning.

Line 597: This github link is broken and different from the one provided in the Declaration section, which works.

Reviewer #2 (Comments for the Author):

This is a very relevant and interesting manuscript describing a method to distinguish living bacteria within a population from dead bacteria or relic DNA. It is well written and designed, with a few minor edits and expansion of discussion required.

Minor Edits:

line 104: italicize *E. coli*

line 120: the Gram status of the two bacteria is written incorrectly.

line 462, 464, 467, 480, 488: all have different versions of x or X – please choose 1 type and make it consistent

line 478 and 547: change min to minutes

line 511 and 521: H₂O should be H₂O

line 462, 477, 530, 544, 548, 555: inconsistent spacing in reference to g following the number. Please make consistent.

line 450 and 483: one should be displayed as 1.

line 426: capitalize t in table 1. Should be Table 1.

line 439, 440, 512, 522: all missing a °C.

line 653: correct ul to µl

line 690: needs space after 5.

line 84: define base pair (bp). Or at next appearance in line 564.

Other:

In lines 168-178 the authors note the presence of spirochetes in the raw vs PMA treated samples and suggest the reads acquired from these samples are relic DNA. The samples are collected from self-reported healthy individuals, however if a more detailed patient history in relation to periodontal status or dental treatments could be provided, that would be very interesting. As the authors noted, these bacteria are more commonly associated with the disease state of periodontitis, however if these are healthy individuals, even the relic DNA is interesting in terms of how long it may reside in the oral cavity. This reviewer appreciates due to the large pool of patient samples, it may be challenging to collect these data. Further discussion on the possibilities or speculation of previous or possibly unknown dental status, such as gingivitis, would be satisfactory instead.

Additionally, the authors should expand the discussion more to include how this method could be very useful to determine the true microbial composition of biofilm communities. Given the amount of extracellular DNA in these environments, relic DNA is likely found in high abundance, particularly in places like the oral cavity. qPCR is a very common method for determining bacterial load, which as the authors suggest, is likely a poor measure for the actual number of live bacteria in this system. Furthermore, this method could be valuable to determine in a more precise manner the ratio of keystone pathogens, such as *Porphyromonas gingivalis* or *Treponema denticola*, which are classically low in abundance but significantly alter the oral microflora in disease, to more common and prolific commensals such as *Streptococcus gordonii*.

Reviewer Comments

Author response

Reviewer comments:

Reviewer #1 (Comments for the Author):

Though there are already a number of studies published out there to validate the stability of the human oral microbiome, this study brings in an interesting perspective with the influence of live cells and total microbial load. These methods should hopefully get adopted by more researchers for better validation of results.

Thank you for this perspective and your thoughtful feedback on our manuscript; as detailed below we have updated the text as recommended.

I only have some minor comments:

Line 120: Fig 1 legend text: should be "Gram positive" and "Gram negative"

We have updated the legend accordingly.

Lines 137 & 139: Should be Fig 2A and Fig 2B; also anywhere else in the document to maintain uniformity with the figure notations

We have capitalized these figure labels.

Line 162, 197, 275: I don't see any supplementary tables submitted in the files section. It would be very interesting and also necessary to have this data made available - particularly the metadata details and the microbial composition files.

Thank you for catching this - due to supplemental file limitations all data tables and metadata files are now publicly accessible through github: [https://github.com/knightlab-analyses/Saliva_quantification_study/tree/master/data](https://github.com/knightlab-analyses/Saliva_quantification_study/tree/master/data)

Line 200: Were there any patterns observed in the salivary loads or composition wrt influence of the type of diet consumed?

We could not identify any patterns between types of food consumed and microbial load or composition – however, this is not completely unexpected due to the relatively low sample size and high diversity of food consumed across participants. The recorded food responses for each participant before sample collection is documented in the metadata file "T1_SMDS_metadata_ms.txt" on the github repository.

Lines 439 & 440: temperature should be "-20 degrees celsius"

Line 536: "...frozen at -20 C" (minus is missing)

Thank you for spotting these omissions - they have been updated accordingly.

Line 542: Any particular rationale for the two sets of samples having been processed differently?
Might be worth mentioning.

The second set of samples was processed in a 96-well plate format to increase the throughput of the lyPMA treatment. This has been clarified in the methods.

Line 597: This github link is broken and different from the one provided in the Declaration section, which works.

Thank you for pointing this out - the github links have been updated and are functional now.

Reviewer #2 (Comments for the Author):

This is a very relevant and interesting manuscript describing a method to distinguish living bacteria within a population from dead bacteria or relic DNA. It is well written and designed, with a few minor edits and expansion of discussion required.

We thank the reviewer for their comments and are grateful for the detailed feedback.

Minor Edits:

line 104: italicize *E. coli*

All bacterial genus and species names have been italicized.

line 120: the Gram status of the two bacteria is written incorrectly.

We have updated this to read 'Gram positive' and 'Gram negative'

line 462, 464, 467, 480, 488: all have different versions of x or X – please choose 1 type and make it consistent

Thank you for catching this; all 'x' are now lower case.

line 478 and 547: change min to minutes line 511 and 521: H₂O should be H₂O

These updates have been made.

line 462, 477, 530, 544, 548, 555: inconsistent spacing in reference to g following the number. Please make consistent.

We have updated all centrifugation units to “x g”.

line 450 and 483: one should be displayed as 1. line 426: capitalize t in table 1. Should be Table 1.
line 439, 440, 512, 522: all missing a °C. line 653: correct ul to µl

line 690: needs space after 5.

line 84: define base pair (bp). Or at next appearance in line 564.

Thank you for your careful reading and catching these typos; we have made corrections accordingly.

Other:

In lines 168-178 the authors note the presence of spirochetes in the raw vs PMA treated samples and suggest the reads acquired from these samples are relic DNA. The samples are collected from self-reported healthy individuals, however if a more detailed patient history in relation to periodontal status or dental treatments could be provided, that would be very interesting. As the authors noted, these bacteria are more commonly associated with the disease state of periodontitis, however if these are healthy individuals, even the relic DNA is interesting in terms of how long it may reside in the oral cavity. This reviewer appreciates due to the large pool of patient samples, it may be challenging to collect these data. Further discussion on the possibilities or speculation of previous or possibly unknown dental status, such as gingivitis, would be satisfactory instead.

Thank you for this comment and we agree that it would be very interesting to compare the relic DNA signals in saliva from healthy versus gingivitis or periodontitis patients in clinically evaluated samples. For example, is the *Treponema* signal less likely to be from relic DNA in patients with gum inflammation or disease? Or does the amount of ‘live’ *Treponema* signal correlate with gum disease progression? It is unfortunate that we were not able to collect more detailed periodontal information on these participants beyond what is self-reported, but we are enthusiastic that this proof-of-concept study will provide the framework to design studies with clinical outcomes in mind.

We have added text to the discussion more explicitly acknowledging these possibilities (lines 282-287)

Additionally, the authors should expand the discussion more to include how this method could be very useful to determine the true microbial composition of biofilm communities. Given the amount of extracellular DNA in these environments, relic DNA is likely found in high abundance, particularly in places like the oral cavity. qPCR is a very common method for determining bacterial load, which as the authors suggest, is likely a poor measure for the actual number of live bacteria in this system. Furthermore, this method could be valuable to determine in a more precise manner the ratio of keystone pathogens, such as *Porphyromonas gingivalis* or *Treponema denticola*, which are classically low in abundance but significantly alter the oral microflora in disease, to more common and prolific commensals such as *Streptococcus gordonii*.

This is an excellent point, as extracellular (or relic) DNA has been shown to serve as scaffolding for biofilms and therefore could be especially misleading when analyzing the sequencing signal. Although it would likely take some optimization, we would be very interested to apply this relic DNA removal technique to oral biofilms, especially dental plaque samples which have more biofilms than saliva. We agree with you that in addition to obscuring the true ratio of keystone pathogens to common commensals, qPCR-based abundance evaluation may very likely be misleading in these biofilm-heavy situations. We have expanded on this in our discussion in lines 294-297.

January 23, 2021

Dr. Karsten Zengler
University of California, San Diego
Bioengineering
9500 Gilman Drive
La Jolla

Re: mSystems01182-20R1 (Quantifying live microbial load in human saliva samples over time reveals stable composition and dynamic load)

Dear Dr. Karsten Zengler:

I am satisfied that the authors have addressed all outstanding reviewer comments, and I am now happy to recommend final acceptance for this manuscript.

Your manuscript has been accepted, and I am forwarding it to the ASM Journals Department for publication. For your reference, ASM Journals' address is given below. Before it can be scheduled for publication, your manuscript will be checked by the mSystems senior production editor, Ellie Ghatineh, to make sure that all elements meet the technical requirements for publication. She will contact you if anything needs to be revised before copyediting and production can begin. Otherwise, you will be notified when your proofs are ready to be viewed.

Sincerely,

Holly Bik
Editor, mSystems

Journals Department
Supplemental Figure 6: Accept
Supplemental Figure 4: Accept
Supplemental Figure 5: Accept
Supplemental Figure 2: Accept
Supplemental Figure 3: Accept
Supplemental Figure 1: Accept
Supplemental Figure 7: Accept